# Human Chorionic Gonadotropin-Stimulated Interleukin-4-Induced-1 (IL4I1) Promotes Human Decidualization via Aryl Hydrocarbon Receptor

**DOI:** 10.3390/ijms24043163

**Published:** 2023-02-05

**Authors:** Jia-Mei Luo, Tong-Tong Zhang, Yu-Ying He, Hui-Na Luo, Yu-Qi Hong, Zeng-Ming Yang

**Affiliations:** 1College of Veterinary Medicine, South China Agricultural University, Guangzhou 510642, China; 2Key Laboratory of Animal Genetics, Breeding and Reproduction in the Plateau Mountain Region, College of Animal Science, Guizhou University, Guiyang 550025, China

**Keywords:** IL4I1, human chorionic gonadotropin, AHR, decidualization, epiregulin

## Abstract

Decidualization is necessary for the successful establishment of early pregnancy in rodents and humans. Disturbed decidualization results in recurrent implantation failure, recurrent spontaneous abortion, and preeclampsia. Tryptophan (Trp), one of the essential amino acids in humans, has a positive effect on mammalian pregnancy. Interleukin 4-induced gene 1 (IL4I1) is a recently identified enzyme that can metabolize L-Trp to activate aryl hydrocarbon receptor (AHR). Although IDO1-catalyzed kynurenine (Kyn) from Trp has been shown to enhance human in vitro decidualization via activating AHR, whether IL4I1-catalyzed metabolites of Trp are involved in human decidualization is still unknown. In our study, human chorionic gonadotropin stimulates IL4I1 expression and secretion from human endometrial epithelial cells through ornithine decarboxylase-induced putrescine production. Either IL4I1-catalyzed indole-3-pyruvic acid (I3P) or its metabolite indole-3-aldehyde (I3A) from Trp is able to induce human in vitro decidualization by activating AHR. As a target gene of AHR, Epiregulin induced by I3P and I3A promotes human in vitro decidualization. Our study indicates that IL4I1-catalyzed metabolites from Trp can enhance human in vitro decidualization through AHR-Epiregulin pathway.

## 1. Introduction

Decidualization is the gradual transformation of endometrial stromal cells into epithelial-like cells and essential to the successful establishment and maintenance of human pregnancy [1]. In mice, implanting embryos invade into endometrium to induce decidualization of uterine stromal cells [2]. Unlike mice, human decidualization spontaneously occurs during the secretory phase of each menstrual cycle regardless of the presence of embryos, and is tightly regulated by the cyclic adenosine monophosphate (cAMP). cAMP plays a critical role in human decidualization primarily by activating protein kinase A and cAMP response element binding protein [3,4]. Although many studies related to decidualization have been reported, the mechanisms underlying the embryo–endometrium interaction are not well understood.

L-tryptophan (Trp), an essential amino acid, can be metabolized to produce serotonin, melatonin, or kynurenine (Kyn) [5,6]. Human placenta can synthesize serotonin [7]. Abnormal serotonin level is associated with gestational diabetic mellitus and preeclampsia [8,9]. Around 95% Trp is metabolized through the Kyn pathway, which is mainly catalyzed by indoleamine-2,3-dioxygenase 1 (IDO1)/tryptophan-2,3-dioxgenase (TDO2) [10]. TDO2 is expressed in the decidualized stromal cells adjacent to the implanted embryo [11]. IDO1 is expressed at the maternal–fetal interface. The dysfunction and abnormal expression of IDO1 are associated with preeclampsia, recurrent spontaneous abortion, and preterm labor [12]. Trp and Kyn metabolism are critical for the establishment and maintenance of pregnancy [13]. Deficiency of Trp or abnormality in Kyn metabolism may lead to recurrent spontaneous abortion, preeclampsia, preterm delivery, and fetal growth restriction [12,14,15].

Kyn can activate aryl hydrocarbon receptor (AHR) [16]. AHR is a ligand-activated cytoplasmic receptor that binds to chaperone proteins or other regulatory proteins to form complexes in the absence of ligands [17,18]. Upon binding to ligands, AHR can translocate into the nucleus and mediate gene regulation [19]. AHR can act as a sensor for a variety of endogenous and exogenous signals essential to differentiation, proliferation and apoptosis, and promote host and microbiome metabolism [20,21]. In the reproductive system, AHR plays an important role in establishing an optimal environment for fertilization, regulating ovarian function and fertility, nourishing the embryo, and maintaining pregnancy [22]. Amphiregulin (AREG) and epiregulin (EREG), members of epidermal growth factor family, are target genes of AHR [23,24]. Meanwhile, AREG and EREG are involved in the embryo development and implantation [25]. However, the regulatory mechanism of AHR on its target genes during human decidualization is still unknown.

IDO1 and TDO2 are highly expressed in cancer cells [26,27]. The activation of Kyn–AHR axis is associated with cancer cell motility [28,29,30]. Because the Kyn–AHR axis can suppress the functions of T cells, targeting on IDO1 inhibition is a promising strategy for restricting the occurrence and development of tumor [31,32]. However, the clinical trial on small molecular inhibitors of IDO1 fails, indicating that AHR may also be activated by alternative pathways [33]. Recent studies have shown that Trp can also be metabolized by interleukin 4-induced gene 1 (IL4I1) [34]. IL4I1 a secreted glycosylated protein, is expressed by antigen-presenting cells including macrophages and dendritic cells, as an enzyme associated with immune regulation [35].

IL4I1 catalyzes the oxidative deamination of phenylalanine, tyrosine and Trp to produce phenylpyruvic acid (PPA), hydroxyphenylpyruvic acid and indole-3-pyruvic acid (I3P), respectively [36]. Among these 3 metabolites, I3P is the only one that can activate AHR [36]. I3P can be further metabolized to produce Indole-3-acetic acid, indole-3-aldehyde (I3A) and indole-3-lactic acid, among which only I3A can activate AHR [36]. Our previous study demonstrated that Trp can promote human in vitro decidualization via activating IDO1-AHR axis [37]. However, whether IL4I1-mediated AHR activation is involved in human decidualization is still unknown.

In this study, we assumed that IL4I1 is involved in human decidualization. The aim of our study was to analyze how IL4I1 from uterine epithelial cells regulates human in vitro decidualization. Our data indicated that human chorionic gonadotropin (HCG) can stimulate IL4I1 expression and secretion from uterine epithelial cells through ornithine decarboxylase 1 (ODC1)-derived putrescine (PUT). I3P and I3A, the IL4I1-catalyzed metabolites of Trp stimulate human decidualization via AHR-EREG signaling.

## 2. Results

### 2.1. Effects of I3P on Human In Vitro Decidualization

IL4I1 catalyzes the oxidative deamination of phenylalanine and Trp to produce PPA and I3P, respectively. PPA is the predominant catalytic product of IL4I1. We first tested the effect of PPA on human in vitro decidualization. Insulin growth factor binding protein 1 (IGFBP1) and prolactin (PRL) are markers of human in vitro decidualization [38]. When stromal cells were treated with 5 μM, 50 μM or 500 μM PPA for 4 days under in vitro decidualization, there was no obvious change on the mRNA levels of *IGFBP1* and *PRL*, demonstrating that PPA shouldn’t promote human in vitro decidualization (Figure 1A,B).

Among the 3 metabolites of IL4I1, I3P is the only one that can activate AHR. When stromal cells were treated with 100 μM or 200 μM I3P for 4 days under in vitro decidualization, the mRNA levels of *IGFBP1* and *PRL* were significantly increased (Figure 1C,D). FOXO1 plays an important role during human decidualization [39]. Under in vitro decidualization, the mRNA and protein levels of FOXO1 were significantly up-regulated by I3P (Figure 1E–G). Stromal-epithelial transition is an important physiological process during decidualization. I3P also promoted the stromal to epithelial transition under in vitro decidualization (Figure 1H). These results indicated that I3P could stimulate human in vitro decidualization.

### 2.2. I3P Stimulates Human In Vitro Decidualization by Activating AHR

Because I3P can activate AHR, we wondered whether I3P stimulated human decidualization via activating AHR. Under in vitro decidualization, nuclear AHR immunofluorescence was obviously increased compared to control. I3P treatment further intensified nuclear AHR fluorescence intensity (Figure 2A). Western blot analysis confirmed the increase of AHR in nuclear fraction (Figure 2B,C). CYP1A1, CYP1B1, and TIPARP are the target genes of AHR. The mRNA levels of *CYP1A1*, *CYP1B1*, and *TIPARP* were significantly increased under in vitro decidualization (Figure 2D), and further upregulated by 200 μM I3P (Figure 2D). I3P-induced the increase of *CYP1A1*, *CYP1B1*, and *TIPARP* mRNA levels was abrogated by treatment with 3 μM CH223191, a specific AHR inhibitor. I3P-stimulated *IGFBP1* expression was also abrogated by CH223191 (Figure 2F). These results indicated that AHR signaling was physiologically activated under human in vitro decidualization and further intensified by I3P.

### 2.3. Effects of I3A on Human In Vitro Decidualization

I3A is a metabolic product of I3P. We tested whether I3A was involved in I3P induction on human decidualization. Under in vitro decidualization, when human stromal cells were treated with 100 μM or 200 μM I3A, there was significantly increased the mRNA levels of *IGFBP1*, *PRL* and *FOXO1* (Figure 3A–C). The protein level of FOXO1 was also increased by I3A (Figure 3D,E). Compared with I3P, I3A-induced increase on *IGFBP1*, *PRL* and *FOXO1* levels was lower. I3A treatment also induced from the stromal to epithelial transition (Figure 3F).

### 2.4. I3A Promotes Human In Vitro Decidualization by Activating AHR

Because I3A is able to activate AHR, we wondered whether I3A induced human decidualization through activating AHR. After stromal cells were treated with I3A for 4 days under in vitro decidualization, nuclear AHR immunofluorescence was obviously increased (Figure 4A). Western blot results also showed that AHR protein level in nuclear fraction was significantly upregulated (Figure 4B,C). Except for *TIPARP*, the mRNA levels of *CYP1A1* and *CYP1B1* were significantly increased after treatment with 200 μM I3A (Figure 4D), which was abrogated by treatment with 3 μM CH223191 for 2 days (Figure 4E). I3A-induced *IGFBP1* increase was also suppressed by CH2231911 (Figure 4F). These results indicated that I3A stimulated human in vitro decidualization by activating AHR.

### 2.5. I3P and I3A Regulate Human In Vitro Decidualization via AHR-EREG Pathway

AREG and EREG are target genes of AHR and play an important role during embryo implantation. We explored whether I3P and I3A promoted human in vitro decidualization through the AHR-AREG/EREG pathway. When stromal cells were treated with different concentrations of I3P, AREG protein levels were significantly up-regulated (Figure 5A,B). I3A treatment also increased AREG protein levels (Figure 5C,D). However, treatment of stromal cells with AREG had no obvious effects on the mRNA level of *IGFBP1* (Figure 5E). Under in vitro decidualization, *EREG* mRNA level was significantly increased, which was further stimulated by I3P treatment (Figure 5F). I3P-induced *EREG* increase under in vitro decidualization was abrogated by CH223191 (Figure 5G). I3A treatment also promoted *EREG* mRNA level under in vitro decidualization (Figure 5H), which was suppressed by CH223191 (Figure 5I). Under in vitro decidualization, EREG could significantly stimulate *IGFBP1* mRNA level (Figure 5J). *IGFBP1* mRNA level under in vitro decidualization was significantly inhibited by CH223191, which could be partially rescued by EREG treatment (Figure 5K). These results showed that I3P and I3A should stimulate human in vitro decidualization through the AHR-EREG pathway.

### 2.6. IL4I1 Expression and Secretion Is Promoted by HCG

During human early pregnancy, HCG is mainly produced by blastocysts and differentiated syncytial trophoblast cells, and is a key embryonic signal necessary for the maintenance of pregnancy. To explore the effect of HCG on IL4I1 levels in human endometrial epithelial cells, endometrial epithelial Ishikawa cells were treated with 0.1, 1 and 10 μg/mL HCG for 1 h, respectively. Compared to control, treatment with 1 μg/mL or 10 μg/mL HCG significantly increased intracellular IL4I1 protein level (Figure 6A,B). IL4I1 is a secreted glycosylated protein. Western blot also confirmed that IL4I1 secretion was obviously increased after Ishikawa cells were treated with 0.1, 1, and 10 μg/mL HCG (Figure 6C). These results suggested that HCG could stimulate IL4I1 expression and secretion in Ishikawa cells.

### 2.7. HCG Regulates IL4I1 Expression through Polyamine Metabolism

In mouse ovary, Odc1 expression and PUT level are stimulated by HCG. When Ishikawa cells were treated with 0.1, 1 and 10 μg/mL HCG for 12 h, ODC1 protein level was significantly increased by 1 and 10 μg/mL HCG, respectively (Figure 7A,B). After Ishikawa cells were treated with 10 μg/mL HCG for 12 h, PUT production was significantly increased (Figure 7C). Treatments of Ishikawa cells with different concentrations of PUT (1 mM and 10 mM) caused a significant increase of intracellular IL4I1 protein level (Figure 7D,E). IL4I1 secretion from Ishikawa cells was also stimulated by 1 or 10 mM PUT (Figure 7F). DFMO, a specific inhibitor of ODC1 [40], was able to abrogated HCG-induced IL4I1 secretion (Figure 7G,H). These results indicated that HCG could stimulate IL4I1 secretion from Ishikawa cells via upregulating ODC1.

## 3. Discussion

Our study first identified that HCG can promote IL4I1 expression and secretion through ODC1-derived PUT in human endometrial epithelial cells. Both I3P and I3A of IL4I1-catalyzed metabolites of Trp can stimulate human in vitro decidualization via activating the AHR-EREG pathway.

Trp is one of the essential amino acids in humans and required for mammalian pregnancy. Trp deprivation of pregnant rats leads to marked dwarfism [41]. The Kyn pathway catalyzed by IDO1/TDO2 is considered as the main Trp catabolic route in humans [32,42]. Ido1-deficient mice exhibit impaired placentation and typical preeclamptic phenotype [43]. In mouse uterus, Tdo2 is mainly expressed in the decidual cells on days 6–8 of pregnancy and can promote mouse in vitro decidualization [44]. During mouse pregnancy, L-Kyn level increases by 3-folds. Kyn lack leads to an increased risk of preeclampsia [45]. Trp and Kyn are conducive to pregnancy adaptation [13,37,45]. Although our previous study indicated that IOD1-metabolized Kyn of Trp can promote human in vitro decidualization through activating AHR signaling [37], whether IL4I1-mediated AHR activation participates in human decidualization is still unknown. I3P and I3A are Trp metabolites catalyzed by IL4I1 and can activate AHR [33]. In our study, either I3P or I3A is able to stimulate human in vitro decidualization through inducing AHR activation.

CYP1A1, CYP1B1 and TIPARP are the target genes of AHR [46,47]. The mRNA levels of these target genes are significantly enhanced by I3P and I3A, which are inhibited by CH223191, a specific AHR inhibitor. I3P- or I3A-induced human in vitro decidualization is also suppressed by CH223191. AHR plays an active role in many aspects of the reproductive system [22,48]. Both 17β-estradiol and progesterone modulate the expression of AHR-associated genes in rat uterus [49,50]. AHR-deficient mice have difficulty in maintaining lactation, rearing pups, and having fewer and smaller litters [51,52].

AREG and EREG are also target genes of AHR [23,24]. In our study, although I3P or I3A stimulates the expression of both AREG and EREG, only EREG can promote in vitro decidualization. Moreover, CH223191 suppression on I3P- or I3A-induced human in vitro decidualization is partially rescued by EREG. This suggests that I3P and I3A promote human in vitro decidualization via AHR-EREG pathway. In rat uterus, AREG is significantly up-regulated by P4 [53]. EREG is expressed in the mouse uterine luminal epithelium and underlying stroma adjacent to the implanting blastocyst [54]. EREG is involved in the embryo development and implantation [25,55]. Our data indicated that I3P and I3A of the Trp metabolites can stimulate human decidualization via AHR-EREG pathway.

Among Trp-degrading enzymes, IL4I1 is more closely associated with AHR activity than IDO1 or TDO2 [36]. IL4I1 is an amino acid oxidase secreted from immune cells and associated with inflammation [56,57]. IL4I1 is expressed by cancer cells in response to IFNγ and TNF-α [58]. IFNγ can promote human in vitro decidualization [37]. TNFα is strongly expressed and secreted by mouse blastocysts, and can promote embryo implantation [59]. During in vitro fertilization, TNFα treatment can increase implantation rate [59,60]. In our study, HCG induces IL4I1 expression and secretion from human endometrial epithelial cells. HCG is strongly secreted by blastocysts and syncytial trophoblast cells during human early pregnancy [61,62]. It is possible that blastocysts-derived HCG stimulates IL4I1 expression and secretion from uterine epithelial cells. A recent study showed that bacterial infection can induce IL4I1 expression [63]. During human embryo implantation, Lactobacillus is the dominated microbiota [64]. It is also possible that uterine microbiota could induce IL4I1 secretion from uterine epithelial cells. Nevertheless, our study first confirmed the presence of endogenous IL4I1 in human endometrial epithelial cells and its metabolites can stimulate human in vitro decidualization.

Polyamines are closely linked to the reproductive processes. Polyamine deficiency induces pregnancy dormancy, PUT supplement is able to terminate the pregnancy dormancy [65]. ODC1, the rate-limiting enzyme for polyamine synthesis, catalyzes the conversion of ornithine to PUT [66]. In mouse uterus, Odc1 is highly expressed in decidual cells [67]. In the mouse, Odc1 inhibition during early pregnancy can significantly block embryo implantation and decidualization [68]. Our data demonstrated that HCG can stimulate IL4I1 expression and secretion from human endometrial cells through activating ODC1 to produce PUT.

In conclusion, our data suggested that HCG promotes IL4I1 expression and secretion by regulating polyamine metabolism in human endometrial epithelial cells. IL4I1-catalyzed I3P and I3A of Trp can stimulate human in vitro decidualization through activating AHR-EREG signaling.

## 4. Materials and Methods

### 4.1. Cell Culture

Human endometrial adenocarcinoma Ishikawa cell line was obtained from Cell Bank of Chinese Academy of Science (Shanghai, China). Ishikawa cells were cultured in phenol red-free Dulbecco’s Modified Eagle’s Medium/Nutrient Mixture F-12 (DMEM/F-12 media, D2906, Sigma-Aldrich, Shanghai, China) supplemented with 10% fetal bovine serum (FBS) (Biological Industries, Cromwell, Israel). Human endometrial stromal CRL-4003 cell line was purchased from American Type Culture Collection (ATCC, Manassas, VA). CRL-4003 cells were cultured with phenol red-free DMEM-F12 containing 1% ITS-G (Gibco, Grand Island, NY, USA) and 10% cFBS (04-0201-1A, Biological Industries, Cromwell, CT, USA). All culture media were supplemented with 100 units/mL penicillin and 0.1 mg/mL streptomycin (Penicillin-Streptomycin, 15140-122, Gibco). Cells were cultured in 5% CO_2_ incubator at 37 °C.

### 4.2. Treatment of Human Endometrial Cells

Human uterine epithelial Ishikawa cells were cultured in DMEM/F12 containing 10% FBS to 95% confluency and further cultured in DMEM/F12 medium with 2% charcoal-treated FBS (cFBS) for 3 h before treatments. Reagents used in treating Ishikawa cells include human chorionic gonadotropin (HCG, hor-250-b, ProSpec-Tany, Rehovot, Israel), putrescine (PUT, P5780, Sigma-Aldrich), indole-3-pyruvic acid (I3P, I7017, Sigma-Aldrich), indole-3-aldehyde (I3A, 129445, Sigma-Aldrich) and difluoromethylornithine (DFMO, D139, Sigma-Aldrich).

Human uterine stromal CRL-4003 cells were cultured in DMEM/F12 supplemented with 10% cFBS. Human in vitro decidualization was induced with 1 μM medroxyprogesterone acetate (MPA, Sigma-Aldrich) and 0.1 mM db-cAMP (Sigma-Aldrich) in DMEM/F12 with 2% cFBS for 4 days as previously described [69]. The media were changed every 2 days.

### 4.3. RNA Extraction and Real-Time PCR

Real-time PCR was performed as previously described [70]. The TRIzol reagent (9109, Takara, Japan) was used for total RNA extraction from cultured cells. The PrimeScript Reverse Transcriptase Kit (Vazyme, Nanjing, Jiangsu, China) was used to reverse transcribe 500 ng of RNA into cDNA from each sample as to the manufacturer’s instructions. For real-time PCR, cDNA was amplified using a SYBR Premix Ex Taq kit (Q311-02-AA, Vazyme) on the CFX96 Touch™ Real-Time System (Bio-Rad, Hercules, CA, USA). The relative changes for the mRNA level of each gene were standardized to *RPL7* mRNA level using the 2^-ΔΔCt^ method. All primer sequences used for real-time RT-RCR in this study were listed in Table 1.

### 4.4. Western Blot

Western blot was performed as previously described [71]. In brief, total proteins of cultured cells were lysed in lysis buffer (150 mM NaCl, 50 mM Tris- HCl, pH 7.5, 5 mM EDTA, 1% Triton X-100, 10 mM NaF, 1 mM Na3VO3, 0.1% SDS, and 1% Sodium deoxycholate) supplemented with protease inhibitor cocktail (4693116001, Roche, Basel, Switzerland). The supernatants collected from cultured epithelial Ishikawa cells with serum-free culture medium were dialyzed in 2 L distilled water for 2 h at room temperature with stirring and lyophilized in a vacuum lyophilizer (FreeZone Plus, LABCONCO, Kansas, MO, USA) for 8 h. The lyophilized powder was dissolved in lysis buffer. The BCA kit (23225, Thermo Scientific, Shanghai, China) were used to measure protein concentration. Protein lysates (5–10 μg) were electrophoresed in SDS-PAGE gels and transferred onto polyvinylidene fluoride (PVDF) membranes (IPVH00010, Millipore, Billerica, MA, USA). Membranes were blocked with 5% non-fat milk (A600669, Sangon, Shanghai, China) for 1 h at room temperature and then incubated with the specific primary antibody overnight at 4 °C. After the membranes were incubated with horseradish peroxidase (HRP)-conjugated secondary antibody (1:5000, Invitrogen, Carlsbad, CA, USA) for 1 h at room temperature, the signals were detected with an ECL Chemiluminescent Kit (WBKLS0100, Millipore, Burlington, MA, USA) on Tanon Imaging System (5200, Tanon, Shanghai, China). The primary antibodies used in this study included anti-Tubulin (#2144, 1:1000, Cell Signaling, Danvers, MA, USA), anti-GAPDH (#sc-32233, 1:1000, Santa Cruz, CA, USA), anti-ODC1 (#28728-1-AP, 1:1000, Proteintech, Rosemont, IL, USA), anti-IL4I1 (#ab222102, 1:1000, Abcam, Cambridge, UK), anti-Integrin β3 (#13166s, 1:1000, Cell Signaling Technology), anti-HOXA10 (#Sc-28620, 1:1000, Santa Cruz), anti-FOXO1 (#2880, 1:1000, Cell Signaling Technology), anti-AREG (#A1860, 1:500, Abclonal, Woburn, MA, USA), anti-Lamin A/C (#2032s, 1:1000, Cell Signaling Technology), anti-AHR (#NB100-2289SS, 1:1000, Novus Biologicals, Englewood, CO, USA).

### 4.5. Nuclear and Cytoplasmic Fractions

Subcellular nuclear-enriched fraction was isolated as previously described [72]. Cultured cells were collected in buffer B (5 mM EDTA in PBS). After centrifugation, the pellets were collected and resuspended in buffer A (10 mM HEPES, 10 mM KCl, 1.5 mM MgCl₂, 1 mM NaF, 1 mM glycerol phosphate with fresh added 2.5% Nonidet P-40), placed on ice for 20 min and vortexed for 15 s. After centrifugation, the supernatant was collected as cytoplasmic fraction. The remaining precipitate was resuspended in buffer C (20 mM HEPES, 420 mM NaCl, 1.5 mM MgCl₂, 1 mM NaF, 1 mM glycerol phosphate with fresh added 2.5% NP-40) and shaken at 4 °C for 25 min. The supernatant was collected as nuclear extract after centrifugation. The protein concentrations of cytoplasmic and nuclear fractions were measured as described above.

### 4.6. Immunofluorescence

Immunofluorescence was performed as previously described with modifications [73]. Briefly, cells cultured on glass coverslips in 24-well plates were fixed with 4% paraformaldehyde in PBS for 15 min, followed by three washing with PBS. After permeabilization with 0.1% Triton X- 100 for 15 min, coverslips were incubated with 10% horse serum for 1 h at 37 ℃ for blocking non-specific binding and incubated with anti-AHR antibody (1:300, 21854-1-AP, Santa Cruz) at 4 °C overnight. After washing 3 times with 1×PBS for 3 min each, coverslips were incubated with a secondary antibody conjugated to FITC. Nuclei were counterstained with 4′6-diamidino-2-phe-nylindole dihydrochloride (DAPI) or propidium iodide (PI) for 1h at 37 ℃. The fluorescent signals were examined under a Leica TCS confocal laser-scanning microscope (Leica, Heidelberg, Germany).

### 4.7. PUT Measurement by ELISA

Ishikawa cells in 6 cm plates were cultured in DMEM/F12 containing 10% FBS to 95% confluency and further cultured in DMEM/F12 with 2% cFBS. Cells were treated with 10 μg/mL HCG for 12 h. After removing the supernatant, the cultured cells were gently scraped from culture plates with a small amount of saline and collected for PUT measurement. The level of intracellular PUT in cultured Ishikawa cells were determined using commercial ELISA kits (BS-E6377H1, Jsbossen, Nanjing, China) according to the manufacturer’s instructions.

### 4.8. Statistical Analysis

All of the experiments were independently repeated at least three times. Data were presented as the mean ± standard deviation (SD) unless stated otherwise. The differences between groups were evaluated by one-way ANOVA followed by a two-tailed unpaired Student’s *t*-test. Statistical significance was indicated as *, *p* < 0.05; **, *p* < 0.01; ***, *p* < 0.001; ****, *p* < 0.0001.

## Figures and Tables

**Figure 1 ijms-24-03163-f001:**
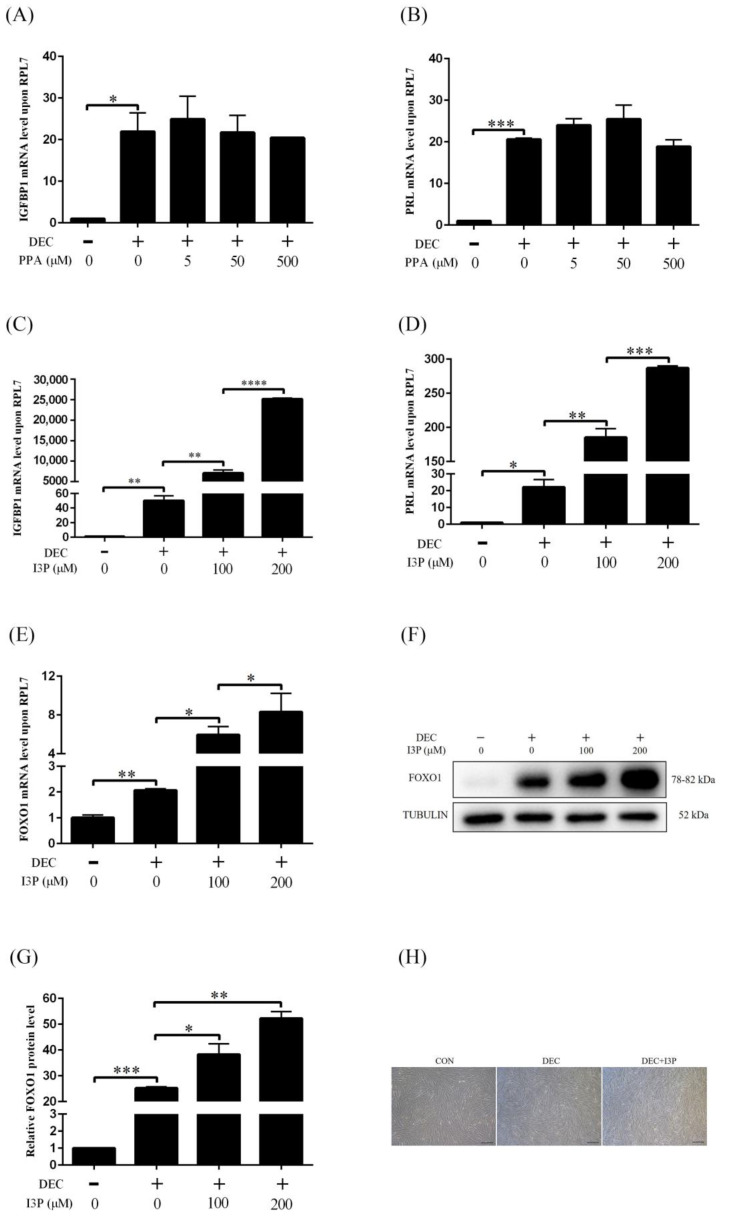
The effects of I3P on human in vitro decidualization. (**A**) RT-qPCR analysis on effects of different concentrations of PPA on *IGFBP1* mRNA level under in vitro decidualization for 4 days. (**B**) Effects of different concentrations of PPA on *PRL* mRNA level under in vitro decidualization for 4 days. (**C**) RT-qPCR analysis on effects of different concentrations of I3P on *IGFBP1* mRNA level under in vitro decidualization for 4 days. (**D**) The effects of different concentrations of I3P on *PRL* mRNA level under in vitro decidualization for 4 days. (**E**) Effects of different concentrations of I3P on *FOXO1* mRNA level under in vitro decidualization for 4 days. (**F**,**G**) Western blot analysis of the protein level of FOXO1. (**H**) Effects of 200 μM I3P on the morphology of human endometrial stromal cells under in vitro decidualization for 4 days. Scale bar = 20 μm. CON, control; DEC, decidualization; PPA, phenylpyruvic acid; I3P, indole-3-pyruvic acid. *, *p* < 0.05; **, *p* < 0.01; ***, *p* < 0.001; ****, *p* < 0.0001.

**Figure 2 ijms-24-03163-f002:**
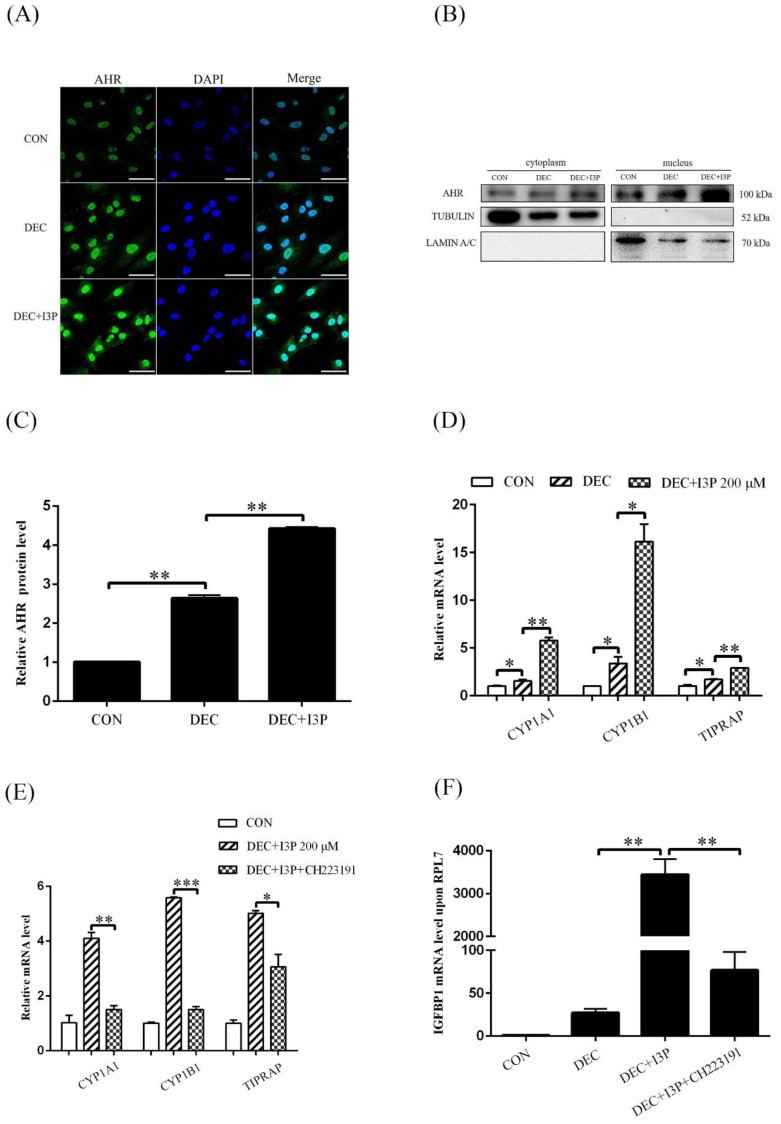
I3P stimulates human in vitro decidualization by activating AHR. (**A**) AHR immunofluorescence of decidual cells treated with 200 μM I3P under in vitro decidualization for 4 days. (**B**,**C**) Western blot analysis on AHR protein level in cytoplasmic and nuclear fractions after stromal cells were treated with I3P under in vitro decidualization for 4 days. (**D**) RT-qPCR analysis on the relative mRNA levels of *CYP1A1*, *CYP1B1* and *TIPARP* after stromal cells were treated with 200 μM I3P under in vitro decidualization. (**E**) RT-qPCR analysis on the mRNA levels of *CYP1A1*, *CYP1B1* and *TIPARP* after stromal cells were treated with I3P and CH223191. (**F**) The mRNA level of *IGFBP1* after stromal cells were treated with I3P and CH223191. Scale bar = 50 μm. CON, control. DEC, decidualization. *, *p* < 0.05; **, *p* < 0.01; ***, *p* < 0.001.

**Figure 3 ijms-24-03163-f003:**
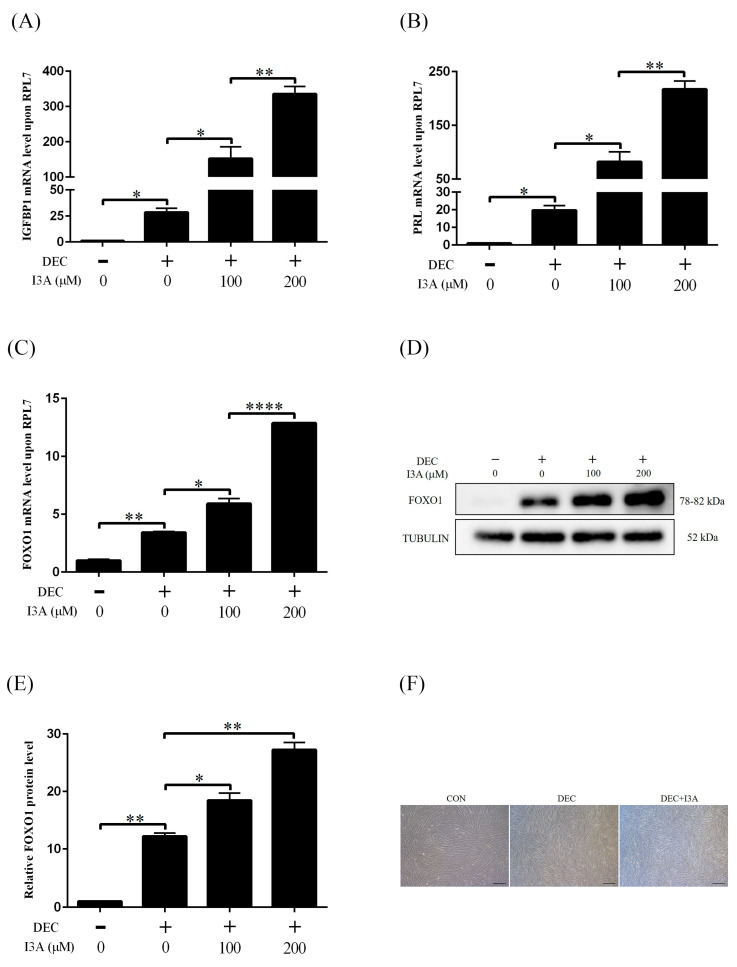
The effects of I3A on human in vitro decidualization. (**A**) RT-qPCR analysis on effects of different concentrations of I3A on *IGFBP1* mRNA level under in vitro decidualization for 4 days. (**B**) The effects of different concentrations of I3A on *PRL* mRNA level under in vitro decidualization for 4 days. (**C**) Effects of different concentrations of I3A on *FOXO1* mRNA level under in vitro decidualization for 4 days. (**D**,**E**) Western blot analysis of the protein level of FOXO1. (**F**) Effects of 200 μM I3A on the morphology of human endometrial stromal cells under in vitro decidualization for 4 days. Scale bar = 20 μm. CON, control. DEC, decidualization; I3A, indole-3-aldehyde. *, *p* < 0.05; **, *p* < 0.01; ****, *p* < 0.0001.

**Figure 4 ijms-24-03163-f004:**
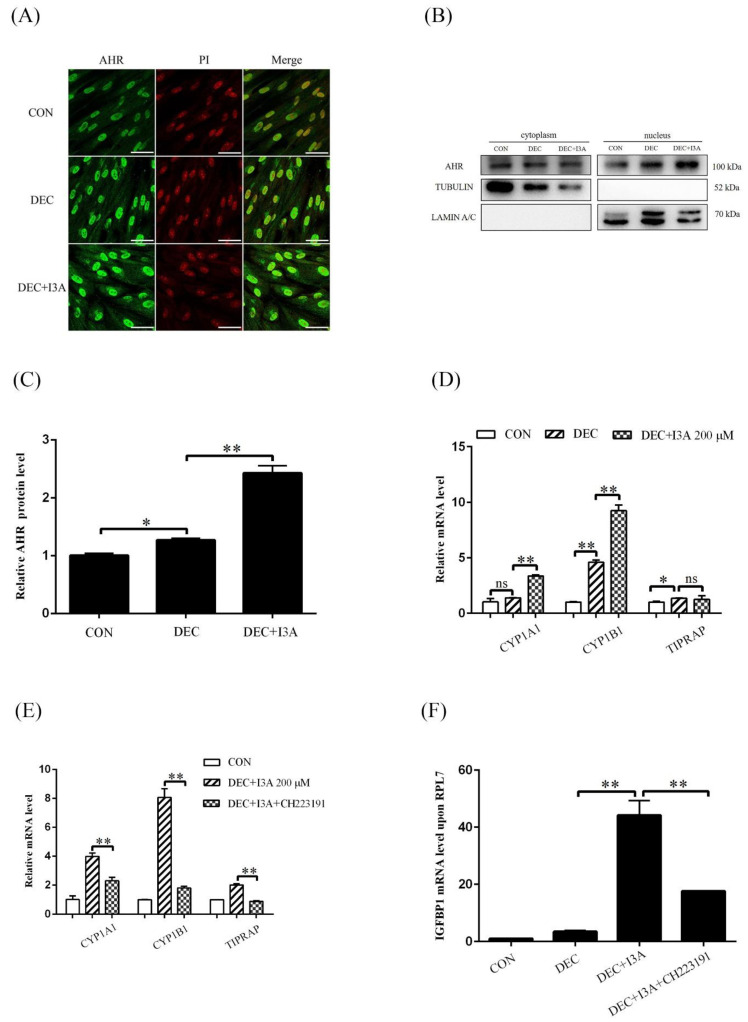
I3A promotes human in vitro decidualization by activating AHR. (**A**) AHR immunofluorescence in decidual cells after stromal cells were treated with 200 mM I3A for 4 days. (**B**,**C**) Western blot analysis of AHR protein level in cytoplasmic and nuclear fractions after stromal cells were treated with I3A for 4 days. (**D**) RT-qPCR analysis on the relative mRNA levels of *CYP1A1*, *CYP1B1* and *TIPARP* after stromal cells were treated with 200 μM I3A. (**E**) RT-qPCR analysis on the mRNA levels of *CYP1A1*, *CYP1B1* and *TIPARP* after stromal cell was treated with I3A and CH223191. (**F**) The mRNA level of *IGFBP1* after stromal cells were treated with I3A and CH223191. Scale bar = 50 μm. CON, control. DEC, decidualization. ns, no significance. *, *p* < 0.05; **, *p* < 0.01.

**Figure 5 ijms-24-03163-f005:**
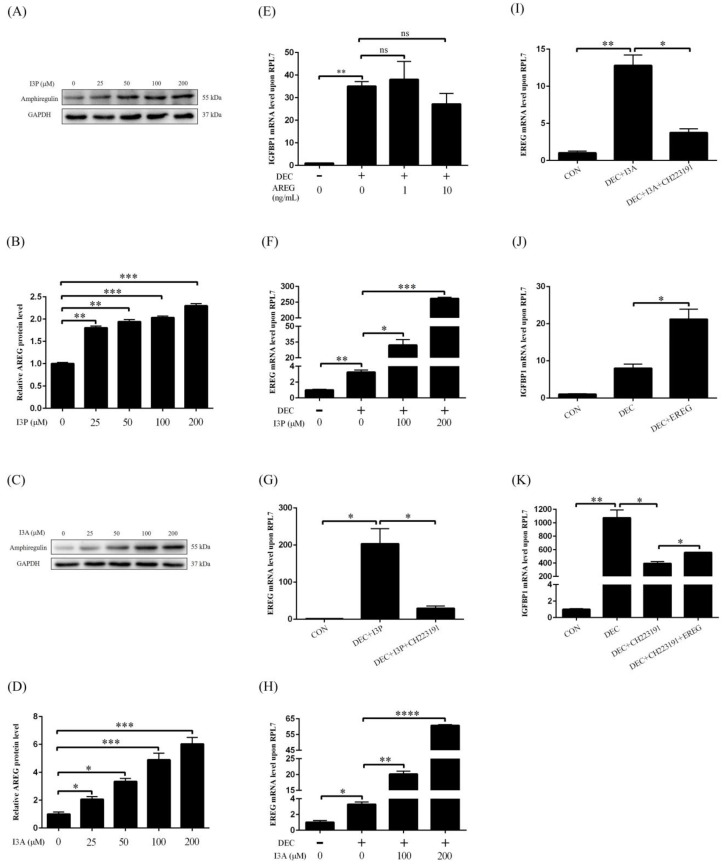
I3P and I3A regulate human in vitro decidualization via the AHR-EREG pathway. (**A**,**B**) AREG protein level after stromal cells were treated with different concentrations of I3P for 24 h. (**C**,**D**) The protein level of AREG after stromal cells were treated with different concentrations of I3A for 24 h. (**E**) RT-qPCR analysis of *IGFBP1* mRNA level after stromal cells were treated with AREG. (**F**) The mRNA level of *EREG* after stromal cells were treated with I3P. (**G**) RT-qPCR analysis of the mRNA level of *EREG* after stromal cells were treated with I3P and CH223191. (**H**) The *EREG* mRNA level after stromal cells were treated with I3A. (**I**) RT-qPCR analysis of the mRNA level of *EREG* after stromal cells were treated with I3A and CH223191. (**J**) *IGFBP1* mRNA level after stromal cells were treated with EREG. (**K**) RT-qPCR analysis of the mRNA level of *IGFBP1* after stromal cells were treated with CH223191 and EREG. CON, control; DEC, decidualization; AREG, amphiregulin; EREG, epiregulin; ns, no significance. *, *p* < 0.05; **, *p* < 0.01; ***, *p* < 0.001; ****, *p* < 0.0001.

**Figure 6 ijms-24-03163-f006:**
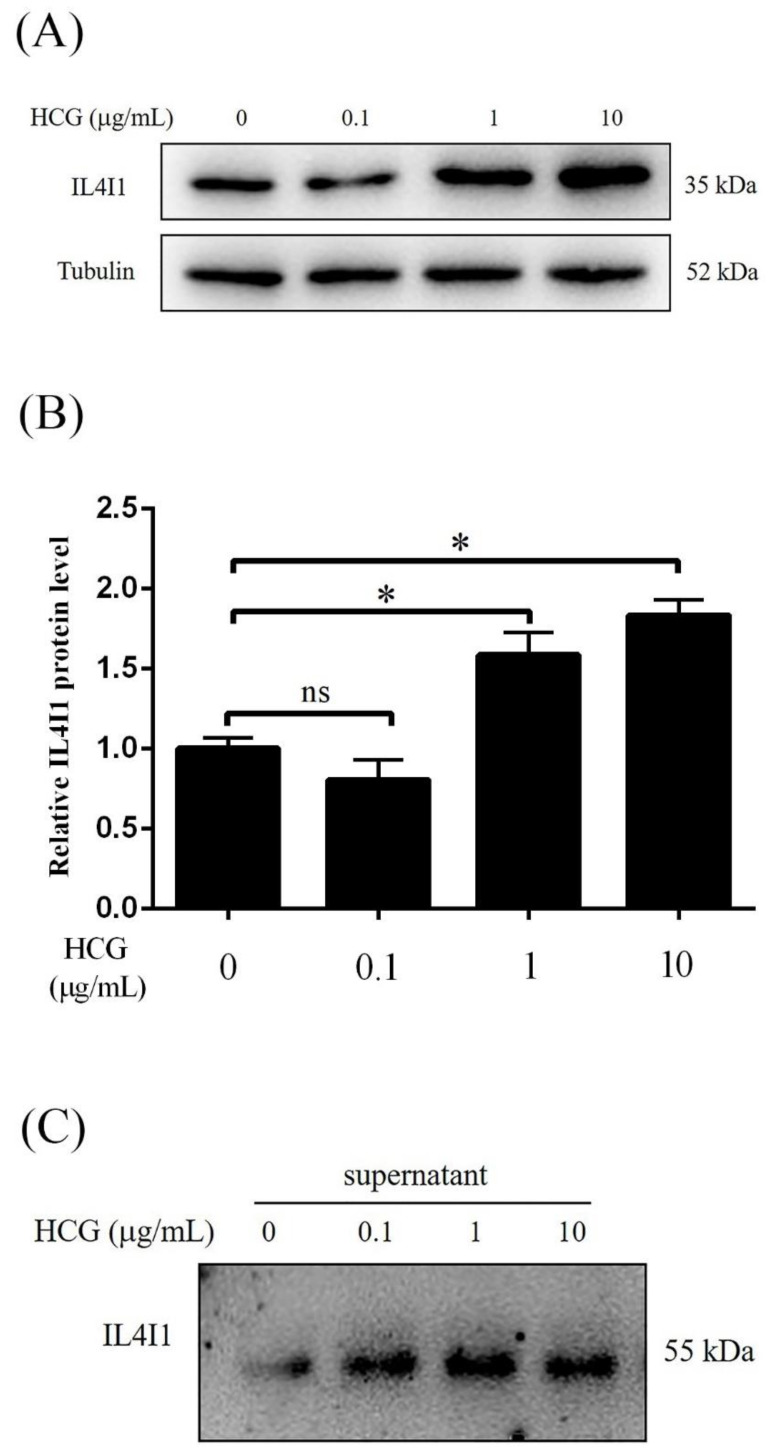
IL4I1 expression and secretion is promoted by HCG. (**A**,**B**) Western blot analysis of effects of HCG on IL4I1 expression in Ishikawa cells. (**C**) The effects of HCG on IL4I1 protein secretion in the supernatant of cultured Ishikawa cells. HCG, human chorionic gonadotropin. ns, no significance. *, *p* < 0.05.

**Figure 7 ijms-24-03163-f007:**
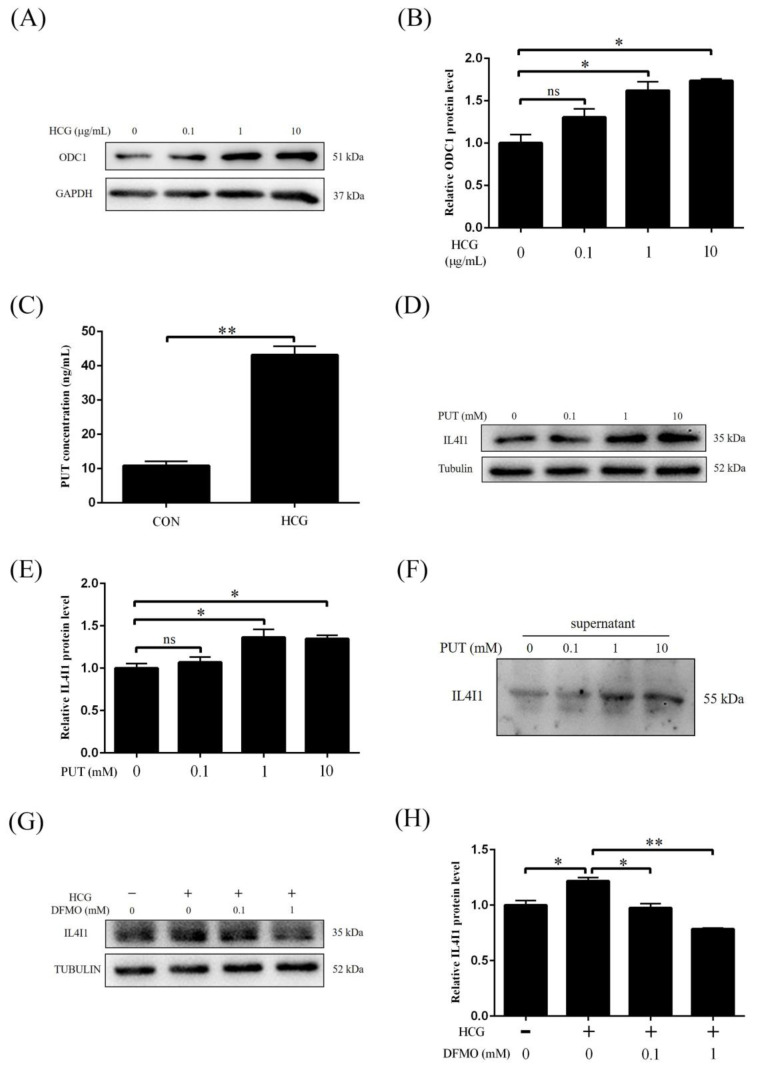
HCG regulates IL4I1 expression through polyamine metabolism. (**A**,**B**) Effects of different concentrations of HCG on the ODC1 protein level in Ishikawa cells. (**C**) PUT concentrations after Ishikawa cells were treated with 10 μg/mL HCG. (**D**,**E**) Western blot analysis of effects of PUT on IL4I1 expression in Ishikawa cells. (**F**) The effects of PUT on IL4I1 protein secretion in the supernatant of cultured Ishikawa cells. (**G**,**H**) Western blot analysis of the protein level of IL4I1 after Ishikawa cells were treated with HCG and DFMO. PUT, putrescine; DFMO, difluoromethylornithine. ns, no significance. *, *p* < 0.05; **, *p* < 0.01.

**Table 1 ijms-24-03163-t001:** Primers for Real-time PCR.

Genes	Primers (5′–3′)	Accession Number	Size (bp)
*CYP1A1* (human)	GTGCGGCAGGGCGATGATTT GGCTGAAGGACATGCTCTGACC	NM_000499.4	85
*CYP1B1* (human)	AACCGCAACTTCAGCAACTT GAGGATAAAGGCGTCCATCA	NM_000104.3	102
*EREG* (human)	GTGTGGCTCAAGTGTCAATAAC GGAACCGACGACTGTGATAAG	NM_001432.3	233
*FOXO1* (human)	CGAGCTGCCAAGAAGAAA TTCGAGGGCGAAATGTAC	NM_00201	105
*IGFBP1* (human)	CCAAACTGCAACAAGAATG GTAGACGCACCAGCAGAG	NM_001,013,029	87
*PRL* (human)	AAGCTGTAGAGATTGAGGAGCAAA TCAGGATGAACCTGGCTGACTA	NM_000948	76
*RPL7* (human)	CTGCTGTGCCAGAAACCCTT TCTTGCCATCCTCGCCAT	NM_000971	194
*TIPARP* (human)	TGAGCCAGACTGTGTAGTGC AACCCCATCAAGTGAGCCAG	NM_001184718.2	193

## Data Availability

Data supporting the findings of this study are available within the paper.

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
