# Peer review of "Human Chorionic Gonadotropin-Stimulated Interleukin-4-Induced-1 (IL4I1) Promotes Human Decidualization via Aryl Hydrocarbon Receptor"

_ijms, 2023, doi:10.3390/ijms24043163_

Round 1

Reviewer 1 Report

The manuscript submitted by Luo et al., entitled 'Human chorionic gonadotropin-stimulated interleukin-4-induced-1 promotes human decidualization via aryl hydrocarbon' reported a novel decidualization mechanism mediated through hCG-induced Odc1 conversion of PUT to stimulate IL4I1 secretion from endometrial epithelial cells that induced trp-induced I3P/I3A stimulated AhR activation of EREG on endometrial stromal cells decidualization. They further demonstrated that DMFO could suppress hCG-induced PUT and IL4I1 expression in epithelial cells, and CH223192 suppressed I3P-induced EREG activation in stromal cells.  Results from this study showed a novel signaling pathway mediated through the hCG-induced AhR-epiregulin pathway on stromal cell decidualization.  I have no primary concern in this study, and the design is solid with convincing results to draw the conclusion. 

Yet, whether IL4I1 could stimulate Trp and IP3 increase needed to be demonstrated in stromal cells, as this is the link between epithelial cells and stromal cell interaction for the decidualization process. Without this, it would be weak to support the conclusion.

Author Response

Olease see the attachment

Reviewer 2 Report

1.      Some grammars and writings should be modified. Here are some examples in the abstract and introduction

EX. in the title, interleukin-4-in-duced-1 promotes human decidualization gene 1 promotes…. ….

EX line 13, kynurenine (Kyn) ftom Trp …..

EX, line 14, metabolites of Trp involved in human in CL along the different stages EX. line 15, human chorionic gonadotropin (HCG) stimulated….

EX. line 17, (ODC1)-induced putrescine production. In this study, some novel proteins were identified along CL development that advance our….

EX. Line 18, indole-3-aldehyde (I3A) from Trp wable to induce human in vitro decidualization by activating AHR.

EX. Line 19-21, epiregulin induced by I3P and I3A, promoted uman n vitro decidualization.

2.      Logistically, the authors need to sow the direct effects of IL4I1 on the decidualization such as by siRNA intervention or enforced expression or pharmacological inhibitors.

Reviewer 3 Report

In the present work, Luo et al. try to explain that human chorionic gonadotropin-stimulated interleukin-4-induced-1 promotes human decidualization via aryl hydrocarbon receptor. There are some questions that should be explained.

1. Editing of English language and style is needed. Pease revise the manuscript throughout.

For example,

Line 9 ‘may’, Line 27 ‘will’, ……, which are not suitable.

2. Abstract,

Please delete (HCG), (ODC1).

‘Epiregulin’ or ‘epiregulin’.

3. Introduction

Line 30, please explain ‘cAMP-PKA-CREB’.

Lines 36-37, please delete ‘Melatonin is beneficial for human embryo development 10. In human assisted reproductive therapy, melatonin has a positive effect on blastocyst rate and pregnancy rate 11.’ Melatonin is not related with this study.

Hypothesis and objective should been added.

4. Materials and methods

Line 86, please explain ‘ATCC’.

Line 90, subscript should be used in ‘CO2’.

Line 112, superscript should be used in ‘2-ΔΔCt’.

Lines 126 and128, ‘for 1 h’ in how many temperature?

5. Results

In general, Results section only presents results, and no discussion pluses citing references. Please revise Results section.

Please add scale bar in all pictures of immunofluorescence in Fig. 2, 4.

6. References, Format of references is not suitable for this Journal.
